# Applications of Metaheuristics Inspired by Nature in a Specific Optimisation Problem of a Postal Distribution Sector

Michał Berliński, Eryk Warchulski [ID] and Stanisław Kozdrowski * [ID]

Institute of Computer Science, Warsaw University of Technology, Nowowiejska 15/19, 00-665 Warsaw, Poland
* Correspondence: stanislaw.kozdrowski@pw.edu.pl; Tel.: +48-22-234-5048

**Abstract:** This paper presents a logistics problem, related to the transport of goods, which can be applied in practice, for example, in postal or courier services. Two mathematical models are presented as problems occurring in a logistics network. The main objective of the optimisation problem presented is to minimise capital resources (Capex), such as cars or containers. Three methods are proposed to solve this problem. The first is a method based on mixed integer programming (MIP) and available through the CPLEX solver. This method is the reference method for us because, as an exact method, it is guaranteed to find the optimal solution as long as the problem is not too large. However, the logistic problem under consideration belongs to the class of NP-complete problems and therefore, for larger problems, i.e., for networks of large size, the MIP method does not find any integer solution in a reasonable computational time. Therefore, two nature-inspired heuristic methods have been proposed. The first is the evolutionary algorithm and the second is the artificial bee colony algorithm. Results indicate that the heuristics methods solve instances of large size, giving suboptimal solutions and therefore, enabling their application to real-life scenarios.

**Keywords:** metaheuristics; evolutionary algorithm; bees algorithm; mixed integer and integer linear programming; combinatorial optimization; logistic problems

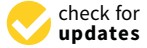



## 1. Introduction

The market for postal, parcel and logistics services requires effective adaptation of the logistics network to product offerings that change with customer expectations. Appropriate and optimal analysis of the impact of designed products on the efficiency of the logistics network and changing parcel streams significantly influences the effectiveness of business decisions in this area, taking into account service-level agreements [1].

Observing the markets for postal and logistics services (parcel services are undoubtedly a common stream for both business profiles), it is possible to notice increasing dynamics of change both on the side of customer requirements and the offer of key players, which is rapidly adapting. The main development trends described, related to the so-called logistics, indicate the necessity to prepare for the introduction of new market and technological solutions on the part of the customers of companies providing postal services, which drives significant changes in the logistics service, and thus in the shape of the network. They result from the growing expectations of customers both with regard to the products themselves and their availability, i.e., consequently, the logistics offer. An operator that wants to remain a key service provider should be prepared to flexibly shape its offer and adapt the shape and potential of the logistics network more and more quickly. In such a dynamic market, historical trend analyses or even real-time observations are no longer sufficient.

Both the level of complexity and the scale of the problem to be solved in postal service companies do not allow the direct use of existing algorithms. Therefore, it is planned to undertake research that will combine the various components of simulation and logistics network optimisation models presented in the literature with specific algorithm properties

that take into account the unique characteristics and complexity (e.g., logistics network, range of products and services) of a postal service company. The optimisation model to be tested is expected to be hybrid in nature, combining a metaheuristic approach with the exact solution of integer programming class subproblems.

This approach proposes a model to describe the optimal service provision in a transport company. The model, presented in mathematical form, proposes a cost function that reflects the minimisation of both capital expenditures, such as cars or containers, and operational expenditures, taking into account human resources, particularly applicable at intermediate nodes for the transfer of goods between cars. The problem belongs to the class of NP-hard problems, so two metaheuristics have been proposed in addition to the exact method, mixed integer programming (MIP), based on integer programming and solved by the CPLEX solver [2]. The exact method, MIP, generally performs well for smaller problems, even giving an optimal solution. For larger logistic networks, on the other hand, it no longer performs so well, failing to give an integer solution within a reasonable computational time. We see MIP as a reference method (benchmark) against metaheuristics. Computation times for larger problems are sometimes several orders of magnitude longer for the MIP method, relative to metaheuristics.

Since the invention of computationally efficient exact algorithms is rather difficult, heuristic discrete optimisation methods seem to be best suited for logistic network optimisation. Therefore, the use of meta-heuristics, designed to be problem-specific, is justified and fully understood, with special attention paid to nature-inspired algorithms with problem operators that have been specifically adapted to the problem under study. In this approach, two nature-inspired algorithms are proposed, i.e., evolutionary algorithms and bee algorithms. The performance of all optimisation algorithms was compared for selected test networks of different complexity.

Considering the practical aspect, it is envisaged that the main areas of analysis supported by the developed approach would be:

- Scenario analysis of strategic decisions in the context of cost optimisation and quality performance of the postal operator's logistics network;
- Modelling the shape of the product portfolio in the context of service opportunities in the logistics network;
- The impact of the planned implementation of new services or acquisition of significant volumes from new customers on the cost position of services in the logistics network problems;
- The possibility of optimising the handling of individual products in line with the changing (increasing) quality requirements of the market, e.g., the possibility of changing the pick-up and delivery times of parcel shipments (of individual product groups) in different regions of the country and the delivery times (of individual product groups) in different regions of the country;
- Analysis of the possibility of bottlenecks in the logistics network with the expected increase in the volumes handled;
- Analysis of the schedules of planned investments in the network of logistics points, taking into account its efficiency parameters;
- Simulation of the demand for transport resources (quantity, type) and the feasibility of transport plans. Selection of optimal development models in the context of the adopted cost and environmental priorities.

From an operations research perspective, delivery operations often boil down to vehicle routing problems that arise when goods need to be delivered to a number of delivery locations by more than one vehicle. This is precisely the kind of problem we consider in this work. Although this broad description covers a wide range of applications, it is often the case that a few specific business characteristics make it difficult to use the approaches associated with this problem, which in turn leads to the development of increasingly specific models that are more applicable to the practical problems faced by businesses.

Over the past decades, the vehicle routing problem (VRP) with its variants has been very popular and addressed in publications by the scientific community [3,4]. In [5], authors examine what is the optimal set of routes for a fleet of vehicles to traverse in order to deliver to a given set of customers. Since VRP is an NP-hard problem [6], exact algorithms are only efficient for small instances of the problem. Heuristics and metaheuristics are often more suitable for practical applications. As real-world problems are much larger in scale (e.g., a company may need to supply thousands of customers from dozens of warehouses, using multiple vehicles and under a variety of constraints).

In the paper [7], the authors used a loading approach for both single and double-decker trucks to deliver goods on multicity routes. The primary objective is to meet both customer demand and delivery times. In [8], the authors investigate what is the shortest possible route for a single vehicle to visit each customer exactly once and returns to the source.

In contrast, in the work [9,10], the authors consider the Container Relocation Problem (CRP), also in maritime transport, using exact methods based on linear programming and heuristics. Containers transported to a container terminal are stored in container yards side by side and on top of each other, forming blocks.

An important aspect of the logistics problems is the consideration of time and delays in the delivery of goods to the point of destination. This has been taken into account in the publications [7,11], where in the latter, a new multi-objective path-finding model is proposed to find optimal paths in road networks with time-dependent stochastic travel times. This study is motivated by the fact that different travellers usually have different route-choice preferences, often involving multiple conflicting criteria, such as expected path travel time, variance of path travel time, and so forth.

In [12], the authors apply the penalty function, where a vehicle is allowed to visit and serve a customer later than their time window, however, a time-dependent late arrival penalty must be taken into consideration if the delayed service occurs.

The proposed approach, as described in this work, includes parts of all the aspects detailed above, i.e., it covers problems such as:

- VRP;
- CRP;
- Time and delays;
- Penalty function.

We propose two metaheuristics based on evolutionary algorithms and the bee swarm algorithm. However, there is no comprehensive approach in the bibliography on this research topic, taking into account all the above aspects, and the objective function represents the cost of the equipment used and the delivery time of the goods, a realistic problem faced by transport, postal and courier companies. The representation of the chromosome in the present case somewhat resembles the one described in the paper in [11], where the chromosome is also used as a vector, with different lengths. However, in our case, the chromosome structure also takes into account the different types of cars. A similar problem has not been considered, and therefore, we cannot compare directly from the literature.

We have previously applied related problems analysed using similar metaheuristics to problems also of the NP-hard type, concerning the design of DWDM optical telecommunication networks [13], and to the knapsack problem, using the CMA-ES algorithm [14].

This paper is organised as follows. In Section 2, the problem is formulated and the logistics service model is described in detail. In Section 3, the proposed algorithms are described in the context of the presented combinatorial optimisation problem. Then, in Section 4, a set of tests for different sizes of the logistics network are presented. The proposed methods are compared and the validity of the metaheuristics used is demonstrated. A statistical and computational efficiency analysis of the proposed methods was performed. Finally, Section 5 presents a summary of the research results and plans for future work on the subject.

## 2. Problem Description

In this section, the logistic problem is formulated using the mixed-integer programming (MIP) method [15]. The sets that are used are shown below, followed by constants and variables. The logistical problem under consideration is divided into two problems and is described by the models presented below.

### 2.1. Basic Logistic Problem

The model of a basic logistic problem (BLP) focuses on the minimisation of the cost function, which is related to the vehicles and the route they travel. The following sets are used in the problem:

$\mathcal{V}$  vertices;

$\mathcal{A}$  arcs;

$\mathcal{S}$  streams;

$\mathcal{T}$  vehicle types;

$\mathcal{M}$  dimensions; $\mathcal{M} = \{weight, pallets\}$;

$\mathcal{F}$  logistic functions; $\mathcal{F} = \{NEN, EDN, PSN\}$; the set is ordered;

$EDN$  logistics function performing activities at a node in the logistics network such as receiving, unloading, loading and dispatch of postal goods;

$PSN$  logistics function performing activities at a node in the logistics network such as processing and sorting postal goods;

$NEN$  logistics function that does not perform any activity in a logistics network node;

$\mathcal{D}$  acceptable delays; $\mathcal{D} = \{D3, D2, D1, D0, DEL\}$; the set is ordered;

$Dx$  means that a package with a posting date $t$ must be delivered by day $t + x$ from the sender to the recipient;

$\delta^+(v)$  set of arcs entering vertex $v \in \mathcal{V}$;

$\delta^-(v)$  set of arcs leaving vertex $v \in \mathcal{V}$.

In the model, the following constants are necessary:

$a(s)$  source of stream $s \in \mathcal{S}$; $a(s) \in \mathcal{V}$;

$b(s)$  destination of stream $s \in \mathcal{S}$; $b(s) \in \mathcal{V}$;

$l(s, m)$  size of stream $s \in \mathcal{S}$ in dimension $m \in \mathcal{M}$;

$d(s)$  maximum acceptable delay of stream $s \in \mathcal{S}$; $d(s) \in \mathcal{D}$;

$\xi(t, a)$  travel cost for vehicle type $t \in \mathcal{T}$ on arc $a \in \mathcal{A}$;

$c(t, m)$  capacity of vehicle type $t \in \mathcal{T}$ in dimension $m \in \mathcal{M}$;

$n(d)$  acceptable delay following acceptable delay $d \in \mathcal{D}$, e.g., $n(D1) = D0$;

$n(f)$  logistic function following logistic function $f \in \mathcal{F}$, e.g., $n(NEN) = EDN$.

Finally, the following variables are used in this problem:

$x_{sadf}$  amount of stream $s \in \mathcal{S}$ with acceptable delay $d \in \mathcal{D}$ after being subjected to logistic function $f \in \mathcal{F}$ on arc $a \in \mathcal{A}$; integer value;

$y_{ta}$  number of vehicles of type $t \in \mathcal{T}$ on arc $a \in \mathcal{A}$; integer value;

$z_{svdf}$  amount of stream $s \in \mathcal{S}$ in vertex $v \in \mathcal{V}$ after being subjected to logistic function $f \in \mathcal{F}$ changing its acceptable delay to $n(d)$; integer value;

$w_{svdf}$  amount of stream $s \in \mathcal{S}$ in vertex $v \in \mathcal{V}$ with acceptable delay $d \in \mathcal{D}$ being subjected to logistic function $f \in \mathcal{F}$; integer value.

The following objective cost function is optimised using an MIP algorithm subject to the constraints listed below.

Objective cost function:

$$\min \sum_{t \in \mathcal{T}} \sum_{a \in \mathcal{A}} y_{ta} \xi(t, a) \tag{1}$$

Constraints:

$$\sum_{a \in \delta^+(v)} x_{sadf} - \sum_{a \in \delta^-(v)} x_{sadf} = z_{svn(d)f} - z_{svdf} + w_{svdn(f)} - w_{svdf} \tag{2}$$

$$\forall s \in \mathcal{S}, \ \forall v \in \mathcal{V}, \ \forall d \in \mathcal{D} \setminus \{DEL\} : d \geq d(s), \ \forall f \in \mathcal{F}$$

$$\sum_{s \in \mathcal{S}} \sum_{d \in \mathcal{D}} \sum_{f \in \mathcal{F}} x_{sadf} l(s, m) \leq \sum_{t \in \mathcal{T}} y_{ta} c(t, m) \quad \forall a \in \mathcal{A}, \ \forall m \in \mathcal{M} \tag{3}$$

$$w_{sa(s)d(s)NEN} = 1 \quad \forall s \in \mathcal{S} \tag{4}$$

$$z_{svDELf} = 0 \quad \forall s \in \mathcal{S}, \ \forall v \in \mathcal{V} \setminus \{b(s)\}, \ \forall f \in \mathcal{F} \tag{5}$$

$$z_{sb(s)DELf} = 0 \quad \forall s \in \mathcal{S}, \ \forall f \in \mathcal{F} \setminus \{PSN\} \tag{6}$$

$$y_{ta} \in \mathbb{Z}^+ \quad \forall t \in \mathcal{T}, \ \forall a \in \mathcal{A} \tag{7}$$

$$x_{sadf} \in \mathbb{Z}^+ \quad \forall s \in \mathcal{S}, \ \forall a \in \mathcal{A}, \ \forall d \in \mathcal{D}, \ \forall f \in \mathcal{F} \tag{8}$$

$$z_{svdf} \in \mathbb{Z}^+ \quad \forall s \in \mathcal{S}, \ \forall v \in \mathcal{V}, \ \forall d \in \mathcal{D}, \ \forall f \in \mathcal{F} \tag{9}$$

$$w_{svdf} \in \mathbb{Z}^+ \quad \forall s \in \mathcal{S}, \ \forall v \in \mathcal{V}, \ \forall d \in \mathcal{D}, \ \forall f \in \mathcal{F}. \tag{10}$$

Objective function (1) serves to minimise the total travel cost. Constraints (2) are the flow conservation constraints. On the left-hand side, we group the traffic entering and leaving a vertex, while on the right-hand side, we group the traffic that either becomes or ceases to be the considered traffic with given $d$ and $f$ by either being kept in a node ($d$ is changing) or being subjected to a logistic function ($f$ is changing). Constraints (3) assure that the adequate number of vehicles is provided on each arc of the network. Constraints (4) assure that each demand enters a network, while constraints (5) and (6) assure that each flow entering a network can leave it only with its destination vertex being completely sorted. In other words, (5) assures it cannot leave a network in a vertex which is not its destination vertex, while (6) assures that it cannot leave a network before being subjected to *PSN* logical function. Constraints (7)–(10) describe the admissible domain of the sets of variables used.

### 2.2. Extended Logistic Problem

The second model we are considering has been named an extended logistic problem (ELP). The model additionally takes into account journey times and associated delays. The following additional constants are used in the ELP model:

Constants:

$\lambda(a)$ travel time on arc $a \in \mathcal{A}$;

$\xi(s, v, f)$ cost for stream $s \in \mathcal{S}$ being subjected to logistic function $f \in \mathcal{F}$ at vertex $v \in \mathcal{V}$;

$\lambda(s, v, f)$ time needed for stream $s \in \mathcal{S}$ to be subjected to logistic function $f \in \mathcal{F}$ at vertex $v \in \mathcal{V}$;

$g(s)$ stream that is gathered together with stream $s$, i.e., both streams have to travel together before being subjected to logistic function $EDN$; $g(s) \in \mathcal{S}$, if $s$ is independent, then $g(s) = s$;

$h(s, d)$ time available to stream $s$ with acceptable delay $d$; e.g., for D+2 stream $s$ available at 7 a.m. that has to be delivered before 4 p.m., we have $h(s, D3) = 0$, $h(s, D2) = 17$,

$$h(s, D1) = 24, h(s, D0) = 16.$$

In this model, the cost function is represented by the expression (11). In addition, two constraints are needed.

Objective cost function:

$$\min \sum_{t \in \mathcal{T}} \sum_{a \in \mathcal{A}} y_{ta} \xi(t, a) + \sum_{s \in \mathcal{S}} \sum_{v \in \mathcal{V}} \sum_{d \in \mathcal{D}} \sum_{f \in \mathcal{F}} w_{svdf} \xi(s, v, f) \tag{11}$$

Constraints:

$$x_{sadNEN} = x_{g(s)adNEN} \quad \forall s \in \mathcal{S}, \ \forall a \in \mathcal{A}, \ \forall d \in \mathcal{D} \tag{12}$$

$$\sum_{a \in \mathcal{A}} \sum_{f \in \mathcal{F}} x_{sadf} \lambda(a) + \sum_{v \in \mathcal{V}} \sum_{f \in \mathcal{F}} w_{svdf} \lambda(s, v, f) \leq h(s, d) \quad \forall s \in \mathcal{S}, \ \forall d \in \mathcal{D}. \tag{13}$$

$$y_{ta} \in \mathbb{Z}^+ \quad \forall t \in \mathcal{T}, \ \forall a \in \mathcal{A} \tag{14}$$

$$x_{sadf} \in \mathbb{Z}^+ \quad \forall s \in \mathcal{S}, \ \forall a \in \mathcal{A}, \ \forall d \in \mathcal{D}, \ \forall f \in \mathcal{F} \tag{15}$$

$$w_{svdf} \in \mathbb{Z}^+ \quad \forall s \in \mathcal{S}, \ \forall v \in \mathcal{V}, \ \forall d \in \mathcal{D}, \ \forall f \in \mathcal{F}. \tag{16}$$

Objective function (11) is to minimise the total travel cost. Constraints (12) assure that streams that were gathered together share their paths before being subjected to logistic function $EDN$. Notice that the relation resulting from (12) is transitive. Therefore, it is possible to gather more than two streams together. Finally, constraints (13) help the model to obtain solutions that are less likely to break real-world delay constraints. They sum up all incurred travel and logistic function times and assure that the sum is below the assigned threshold. Notice that these constraints assume that the throughput of each vertex is infinite; thus, streams do not have to wait before being served. Moreover, they consider weighted averages of incurred delays when streams are split. Constraints (14)–(16) describe the admissible domain of the sets of variables used.

## 3. Methods

We have employed two nature-inspired metaheuristic algorithms, i.e., the evolutionary algorithm (EA) and the artificial bees colony algorithm (ABC), to solve the given optimization problem.

The used methods represent population-based optimisation algorithms and so-called swarm intelligence, which relies on the cooperation of the collective members. The evolutionary algorithm was applied in the most straightforward setting, i.e., (1 + 1), where the population consists of only one individual (a parent), and the offspring is created by adding a random perturbation to the parent. The artificial bee colony algorithm used in our research is consistent with the version of the ABC proposed in [16].

In further subsections, the representation of the individuals and variation operators are described in detail. Additionally, we provide the pseudo-code for each employed metaheuristic method.

### 3.1. Representation

For both metaheuristics, each individual $\mathbf{x_i}$ in population $P = \{\mathbf{x_1}, \ldots, \mathbf{x_k}\}$ is represented as a vector of $m$ vehicles:

$$\mathbf{x_i} = [\mathbf{v_1}, \ldots, \mathbf{v_m}]$$

For the ABC algorithm, the representation is extended with a type of a bee, i.e., employed, onlooker or scout.

A vehicle is represented as a triple:

$$\mathbf{v_i} = (\mathbf{o}, \mathbf{r}, \mathbf{c})$$

where:

- **o** represents a vector of taken objects, i.e., stream realizations (both models);
- **r** represents a vector of a drawn route from the set of prepared paths between all vertices (both models);
- **c** represents capacity $c(t, m)$ of a vehicle on each arc $a \in \mathcal{A}$ of the processed route (both models).

### 3.2. Variation Operators

Both ABC and EA algorithms modify their solutions by using following variation operators:

- `ReplaceCarsOperator` during each invocation removes the number of cars set by the parameter and releases streams that they were transporting. Then, it attempts to fill already existing cars with free streams. If there are any remaining streams left, new cars are generated for them;
- `ExchangeRealisationOperator` removes a single stream from the number of cars set by the parameter. Then, it attempts to randomly fill already existing cars with free streams. If there are any remaining streams left, new cars are generated for them;
- `MixRealisationsOperator` unloads all cars and then it attempts to fill already existing cars with free streams. If there are any remaining streams left, new cars are generated for them.

By default, the probabilities of using operators are as follows:

- `ReplaceCarsOperator`—40%;
- `ExchangeRealisationOperator`—50%;
- `MixRealisationsOperator`—10%.

Both `ReplaceCarsOperator` and `ExchangeRealisationOperator` take two input arguments: $c$ and $r$. The first one determines minimal part of cars/streams modified during a single call, the second one specifies maximal part. During each call, the operators randomize a number from the range given by these parameters.

### 3.3. Evolutionary Algorithm

In this approach, EA (1 + 1) is applied. In each iteration, a new individual is created from the parent by applying a mutation (one of three variation operators, presented in Section 3.2). Then, it is evaluated. If its objective function value is better than its predecessor, then it becomes the base specimen of the next iteration. Otherwise, it is rejected. The EA pseudocode is presented in Algorithm 1. The algorithm takes as parameters the following: $i$—the number of iterations, which is the stop criterion of the algorithm at the same time and mutation operation parameters $c$ and $r$, described in Section 3.2.

---

**Algorithm 1:** Evolutionary Algorithm

---

**Input**: i, c, r
**Output**: Best solution found
Specimen ← RandomSolution
stopCounter ← 0
**while** *Stop condition is not met* **do**
  SpecimenBis ← Specimen
  mutation(SpecimenBis)
  evaluateSpecimens()
  **if** *SpecimenBis is better* **then**
    Specimen ← SpecimenBis
    stopCounter ← 0
  **else**
    stopCounter++

---

### 3.4. Artificial Bee Colony Algorithm

In the case of the ABC algorithm, the first population of $N$ individuals is randomly generated by creating vehicles and assigning streams to them as long as there are any free streams. Then, each individual one is evaluated and depending on its quality, it receives more or fewer onlookers. Onlookers are created by applying one of the variation operators (using $c$ and $r$ parameters) to the worker bees. Then, all the onlookers are evaluated. If an onlooker is better than the worker bee that it is assigned to, the worker is replaced. The best worker bee is stored. Other individuals are rejected after existing for $m$ iterations without improving their fitness function. The algorithm stops after $i$ iterations. The ABC pseudocode is presented in Algorithm 2.

---

**Algorithm 2:** Artificial Bee Colony Algorithm

**Input**: N, m, i, c, r
**Output**: Best solution found
bees ← RandomSolutions(N);
**while** *Stop condition is not met* **do**
    evaluateBees()
    sendOnlookers()
    **switch** *drawnOperator* **do**
        **case** *1* **do**
           └ replaceCarsOperator(cf)
        **case** *2* **do**
           └ exchangeRealisationOperator(rf)
        **case** *3* **do**
           └ mixRealisationsOperator()
    calculateCosts()
    replaceWorkers()
    selectBestBee(bees);

---

## 4. Results

In the ABC algorithm, the following parameters can be specified:

- Size of the workers population—30;
- Number of iterations—17000;
- Maximum number of iterations without improvement for a single bee—40;
- Weight of penalty factor that is penalizing not completely full cars—16.6%.

We have used various population sizes from $n = 15$ to 100, and found that in most cases, it is sufficient to use $n = 15$ to 50. Therefore, we have used population a size of $n = 30$ in all our simulations. We tested the number of iterations in the range (500–2500), and the value of 1700 turned out to be the best compromise between the computation time and the quality of the obtained solution. Other parameters were also selected through empirical research.

The Evolutionary Algorithm worked for 50,000 iterations. The total number of objective function calculations was then almost the same for both tested algorithms. After a series of experiments, `ReplaceCarsOperator` and `ExchangeRealisationOperator` parameters values were determined as:

- Minimal part of cars/streams modified during a single call—6.25%;
- Maximal part of cars/streams modified during a single call—12.5%.

The calculations were performed using a linear solver engine of CPLEX 12.8.0.0 on a 2.1 GHz Xeon E7-4830 v.3 processor with 256 GB RAM, running under the Linux Debian operating system.

Figures 1 and 2 show the convergence curves of the considered algorithms for BLP and ELP models, respectively. Each curve represents an average of 20 independent runs. In

both cases, logistic networks with `NetworkSize` = 5, 10, 15, 20, 25 and 30 were considered. The following observations can be made:

- For the BLP model (Figure 1), the MIP method found a solution in every case, with only `NetworkSize` = 5 being the optimal solution;
- In the case of the ELP model (Figure 2), the MIP method only found an optimum for the five-node network. However, for `NetworkSize` = 20, 25 and 30, it did not find an integer solution at all;
- Both heuristic methods found a sub-optimal solution in every case considered;
- In most cases, EA gives better results. However, for smaller network instances, the ABC method is comparable or slightly better. Such observation is strengthened by the results of the Mann–Whitney test (column T in Tables 1 and 3). In most cases, for both problem variants, there is a significant difference in a location shift in favour of the EA algorithm.

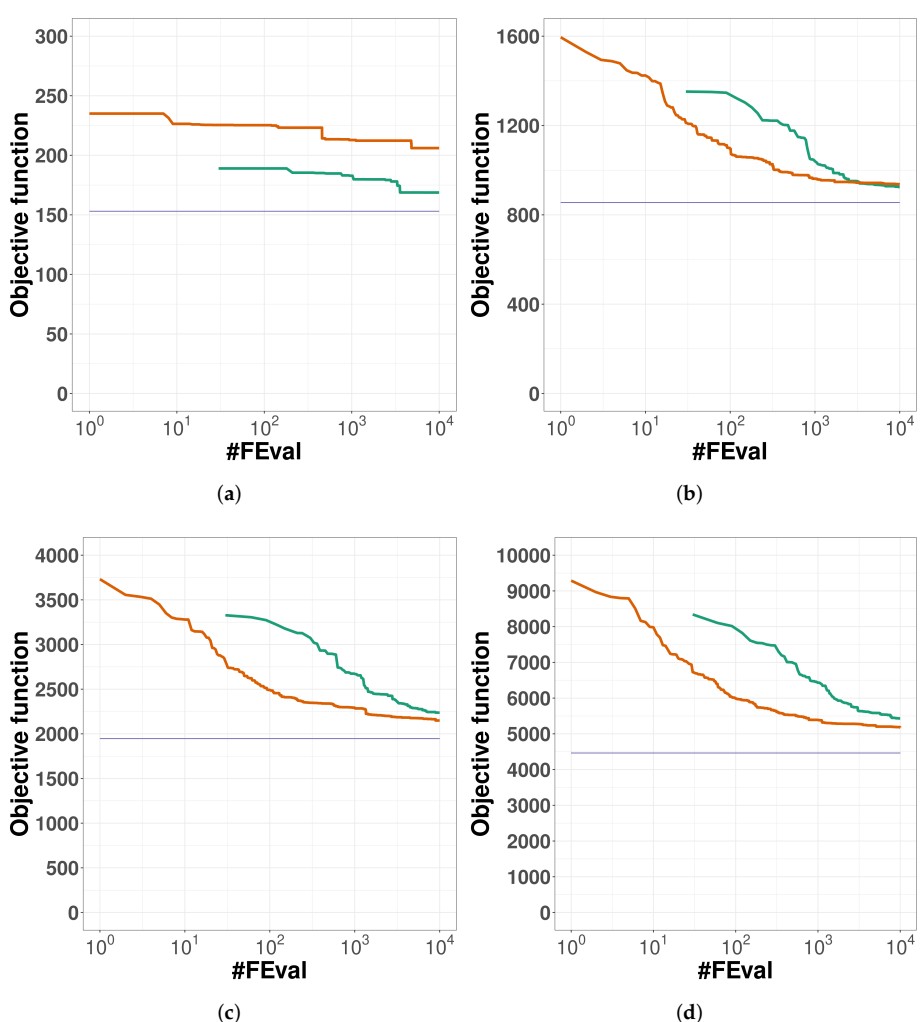

**Figure 1.** *Cont.*

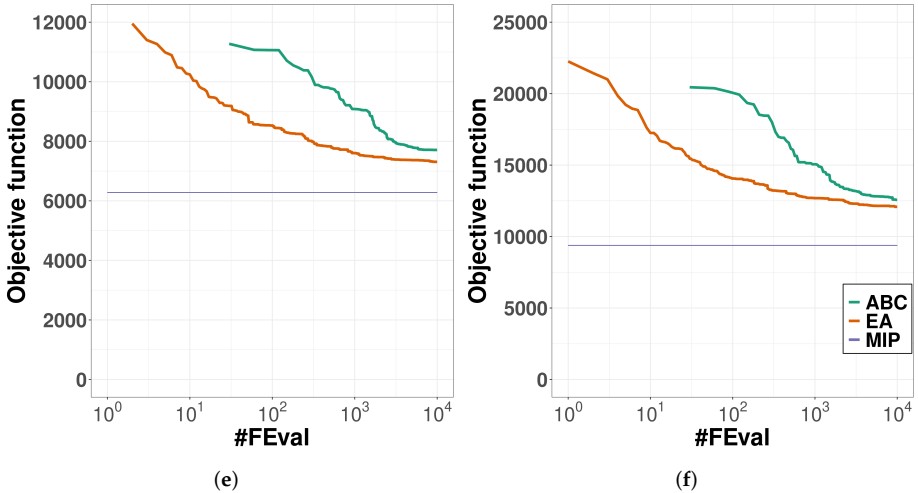

(**e**)                    (**f**)

**Figure 1.** Convergence curves plotted for the algorithms considered, for a logistic network of different sizes, for the BLP model. Each curve is an average of 20 independent runs. (**a**) 5-node. (**b**) 10-node. (**c**) 15-node. (**d**) 20-node. (**e**) 25-node. (**f**) 30-node.

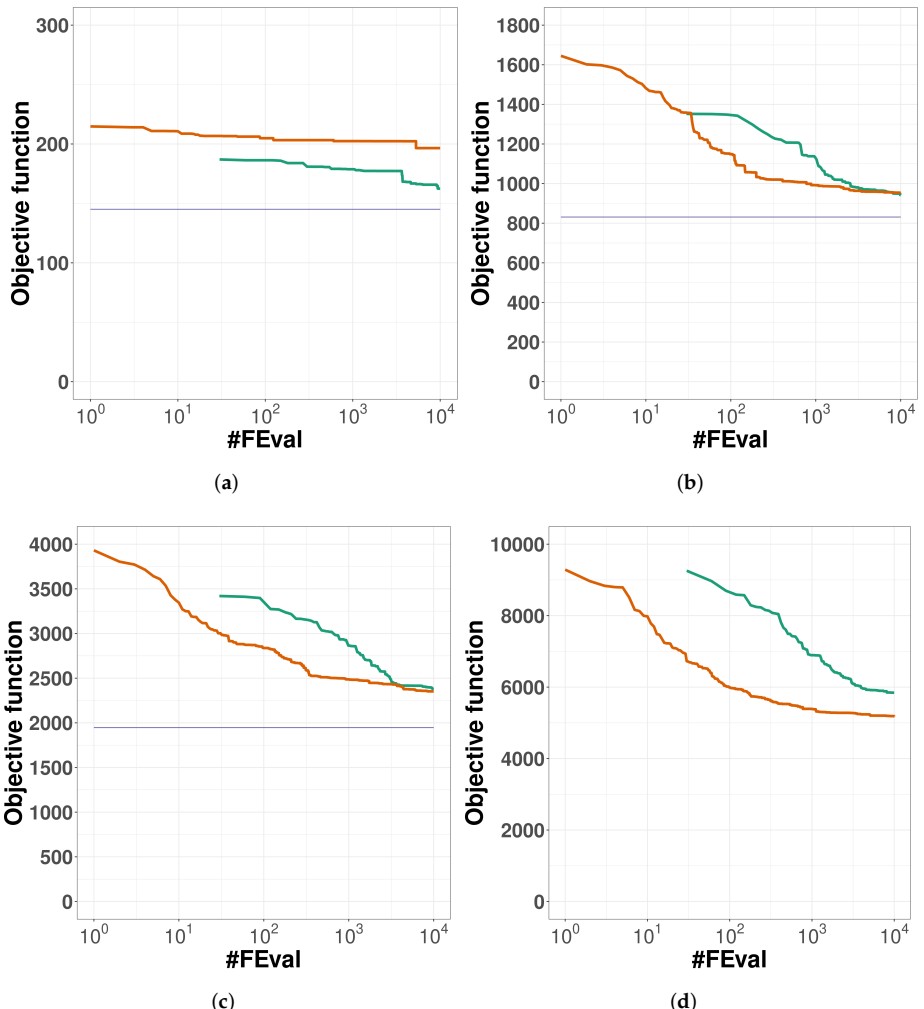

(**a**)                    (**b**)

(**c**)                    (**d**)

**Figure 2.** *Cont.*

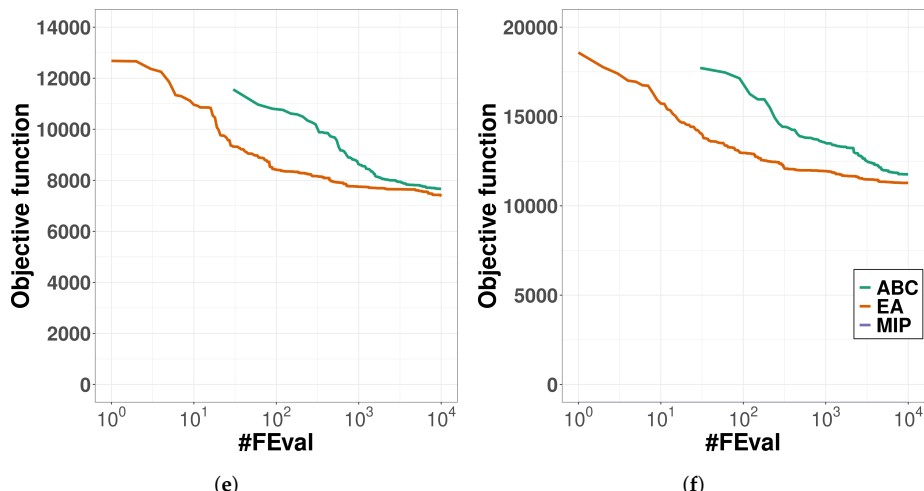

**Figure 2.** Convergence curves plotted for the algorithms considered, for a logistic network of different sizes, for the ELP model. Each curve is an average of 20 independent runs. (**a**) 5-node. (**b**) 10-node. (**c**) 15-node. (**d**) 20-node. (**e**) 25-node. (**f**) 30-node.

**Table 1.** Comparison of the average and the best value of the objective function evaluation for each test case together with the outcomes of the paired Wilcoxon test, for the BLP model. Symbols $+$, $-$ and $=$, in the last column, indicate whether the EA algorithm was better ($+$), worse ($-$) or equal ($=$) to the ABC algorithm.

| Network #Node | EA Mean | EA Best | ABC Mean | ABC Best | MIP | Wilcoxon–Mann–Whitney Test |
|---|---|---|---|---|---|---|
| 5 | 206 | 206 | 167 | 165 | 153 | $-$ |
| 7 | 402 | 401 | 362 | 354 | 341 | $-$ |
| 10 | 935 | 930 | 923 | 911 | 855 | $=$ |
| 12 | 1296 | 1284 | 1325 | 1293 | 1156 | $+$ |
| 15 | 2135 | 2104 | 2241 | 2191 | 1947 | $+$ |
| 17 | 2746 | 2688 | 2850 | 2762 | 2489 | $+$ |
| 20 | 4134 | 4080 | 4201 | 4054 | 3601 | $+$ |
| 22 | 5134 | 5170 | 5111 | 5187 | 4464 | $+$ |
| 25 | 7182 | 7262 | 7154 | 7331 | 6280 | $=$ |
| 30 | 11,378 | 11,260 | 11,331 | 11,113 | 9379 | $=$ |

Tables 1 and 3 show the mean and best values of the objective function in each test case. Whereas, Tables 2 and 4 show the mean and standard deviation values of the computational time in each test case. The last column in each table contains the result of the Wilcoxon–Mann–Whitney test [17] with a confidence level of 0.95 applied to the outcomes of the two metaheuristics (EA and ABC). We used symbols $+$, $-$ and $=$ to indicate whether the EA algorithm was—respectively—better, worse or equal to the ABC algorithm.

Figure 3a,b depict the calculation time achieved by heuristics and the MIP solvers before termination for networks with 5, 10, 20 and 30 nodes for BLP and ELP variants. One may notice that for the BLP variant, the MIP solver on small networks (`NetworkSize` = 5 and 15) outperformed the ABC algorithm and was slightly worse than the EA. However, for the ELP variant, the MIP achieved a comparable calculation time to the heuristics on the network only with five nodes. For each bigger network, i.e., with 20 or 30 nodes, the MIP solver reached the total calculation time stop condition without a feasible solution, while the heuristics returned a suboptimal albeit feasible solution in total calculation time smaller by a factor of 1 or 2 than the MIP runtime. More detailed data are contained in the Tables 2 and 4 together with the results of the Mann–Whitney test. According to the Mann–Whitney test results (column **T**), the EA algorithm was worse than the ABC algorithm only for `NetworkSize` = 5, 7 for BLP and ELP variants. For each bigger size of network

considered in conducted experiments, the EA outperformed the ABC heuristic. The bigger the size of the network, the more noticeable the difference between heuristics. It is still worth emphasising that, despite the fact that the proposed heuristics do not find an optimal solution, suboptimal solutions are acceptable in the practical problems considered, especially as the calculations are not so time-consuming.

**Table 2.** Comparison of the computation time (sec.) for the BLP model together with the outcomes of the paired Wilcoxon test. Symbols +, − and = indicate whether the EA algorithm was better (+), worse (−) or equal (=) to the ABC algorithm.

| Network #Node | EA | | ABC | | MIP | Wilcoxon–Mann–Whitney Test |
| --- | --- | --- | --- | --- | --- | --- |
| | Mean | std | Mean | std | | |
| 5 | 12 | 0.3 | 1.98 | 0.1 | 4 | − |
| 7 | 26 | 1.3 | 7 | 0.3 | 83 | − |
| 10 | 83 | 1.1 | 129 | 8.2 | 83 | + |
| 12 | 132 | 1.54 | 162 | 16.4 | 43,200 | + |
| 15 | 325 | 3.35 | 518 | 37.5 | 43,200 | + |
| 17 | 463 | 10 | 773 | 53.1 | 43,200 | + |
| 20 | 1023 | 22 | 1816 | 155 | 43,200 | + |
| 22 | 1392 | 16 | 2568 | 230 | 43,200 | + |
| 25 | 2813 | 44.4 | 5353 | 298 | 43,200 | + |
| 30 | 5885 | 84 | 12,146 | 1949 | 43,200 | + |

**Table 3.** Comparison of the average and the best value of the objective function evaluation for each test case together with the outcomes of the paired Wilcoxon test, for ELP model. Symbols +, − and = indicate whether the EA algorithm was better (+), worse (−) or equal (=) to the ABC algorithm.

| Network #Node | EA | | ABC | | MIP | Wilcoxon–Mann–Whitney Test |
| --- | --- | --- | --- | --- | --- | --- |
| | Mean | Best | Mean | Best | | |
| 5 | 155 | 151 | 162 | 158 | 145 | + |
| 7 | 342 | 331 | 355 | 350 | 318 | + |
| 10 | 908 | 897 | 942 | 926 | 831 | + |
| 12 | 1308 | 1287 | 1369 | 1341 | 1189 | + |
| 15 | 2162 | 2134 | 2355 | 2281 | 1945 | + |
| 17 | 2808 | 2779 | 2924 | 2873 | 2507 | + |
| 20 | 4229 | 4125 | 4208 | 4055 | − − | = |
| 22 | 5520 | 5391 | 5797 | 5655 | − − | + |
| 25 | 7073 | 7030 | 7519 | 7286 | − − | + |
| 30 | 10,928 | 10,842 | 10,947 | 10,877 | − − | = |

**Table 4.** Comparison of the computation time (sec.) together with the outcomes of the paired Wilcoxon test, for the ELP model. Symbols +, − and = indicate whether the EA algorithm was better (+), worse (−) or equal (=) to the ABC algorithm.

| Network #Node | EA | | ABC | | MIP | Wilcoxon–Mann–Whitney Test |
| --- | --- | --- | --- | --- | --- | --- |
| | Mean | std | Mean | std | | |
| 5 | 14 | 0.2 | 2 | 0.1 | 11 | − |
| 7 | 26 | 0.9 | 7 | 0.2 | 121 | − |
| 10 | 76 | 1.93 | 75 | 11 | 43,200 | = |
| 12 | 132 | 3.6 | 161 | 13 | 43,200 | + |
| 15 | 317 | 4.7 | 533 | 23 | 43,200 | + |
| 17 | 440 | 8.4 | 682 | 45 | 43,200 | + |
| 20 | 1355 | 60 | 2580 | 149 | 43,200 | + |
| 22 | 1355 | 60 | 2588 | 150 | 43,200 | + |
| 25 | 2281 | 80 | 3908 | 528 | 43,200 | + |
| 30 | 4963 | 86 | 11,212 | 1213 | 43,200 | + |

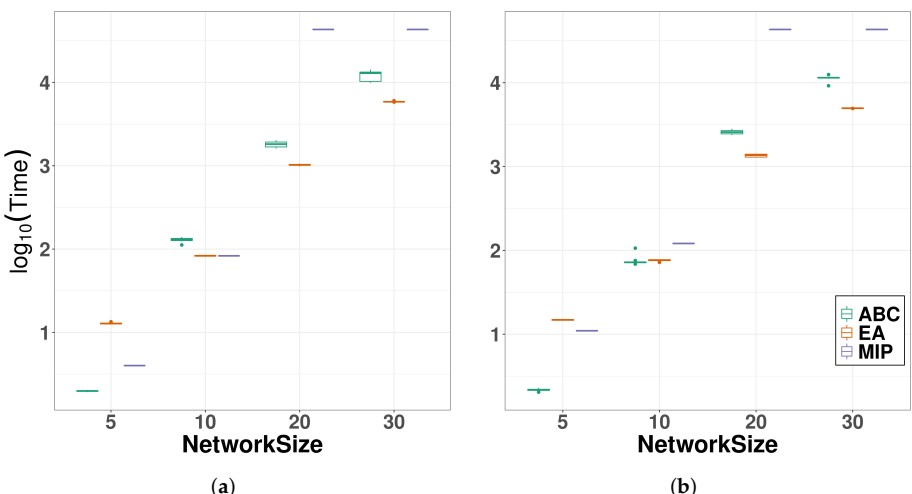

**Figure 3.** Box plot of calculation time for heuristic methods for the BLP—(**a**) and ELP models—(**b**) model.

## 5. Conclusions

This paper presents two models to describe realistic problems in a logistic network. For the BPL problem, the MIP method finds an optimal solution for small network instances and is competitive with the presented heuristic methods. For large network instances, the computation time increases rapidly and the MIP method provides a suboptimal solution needing a rather large computation time. The EA and ABC heuristic methods require, on average, at least an order of magnitude less computation time, although the suboptimal result is approx. 5–10% worse in relation to the exact MIP method. In the case of the model extended by the ELP time parameters, for small problem instances, the MIP method is still 2–3% better than the heuristics. On the other hand, for medium and large problem instances, the exact MIP method does not find an integer solution at all in a reasonable time, while the heuristics give a satisfactory suboptimal integer solution in an acceptable computational time.

Finally, it is worth noting that the results apply to a logistic network of practical importance and therefore, provide additional guidance for network operators who are planning to expand the logistic network, and that the proposed heuristic algorithms give suboptimal results, especially for a more difficult problem with more than 20 nodes. It should be noted that there is still potential for improving the results obtained by the heuristics and this will be our goal in future research work in this area.

**Author Contributions:** Conceptualization, S.K. and E.W.; methodology, S.K. and E.W.; software, M.B. and E.W.; validation, S.K. and E.W.; formal analysis, S.K. and E.W.; investigation, S.K., E.W. and M.B.; data curation, S.K. and M.B.; writing—original draft preparation, S.K., E.W. and M.B.; writing—review and editing, S.K. and E.W.; visualisation, S.K. and E.W.; All authors have read and agreed to the published version of the manuscript.

**Funding:** This research was funded by the National Center for Research and Development under grant POIR.04.01.04-00-0054/17-00). LAS Project under the title Simulation and analysis methods of logistics networks for postal operators. Funding of The National Centre for Research and Development in Poland.

**Institutional Review Board Statement:** Not applicable.

**Informed Consent Statement:** Not applicable.

**Data Availability Statement:** Data can be found at: https://github.com/ewarchul/applsci-2022 /tree/main/src/ampl.

**Conflicts of Interest:** The authors declare no conflict of interest.

## Abbreviations

The following abbreviations are used in this manuscript:

| | |
|---|---|
| EA | Evolutionary Algorithm |
| ABC | Artificial Bee Colony |
| ES | Evolution Strategy |
| MIP | Mixed Integer Programming |
| CAPEX | Capital Expenditure |
| OPEX | Operational Expenditure |
| VRP | Vehicle Routing Problem |
| NP-hard | Non Polynomial |
| CRP | Container Relocation Problem |
| EDN | Expedition and Dispaching Node |
| PSN | Processing and Sorting Node |
| NEN | None Expedition and Processing Node |
| BLP | Basic Logistic Problem |
| ELP | Extended Logistic Problem |
| NSGA | Non-Dominated Sorting Genetic Algorithm |
| gap | difference between current best integer solution and optimal value of LP relaxation |
| CPLEX | Mixed Integer Programming solver |
| CMA-ES | Covariance Matrix Adaptation Evolution Strategy |
| DWDM | Dense Wavelength Division Multiplexing |

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
