# Peer review of "Applications of Metaheuristics Inspired by Nature in a Specific Optimisation Problem of a Postal Distribution Sector"

_applsci, doi:10.3390/app12189384_

Round 1
Reviewer 1 Report
ID: applsci-1888752-peer-review-v2
Title: Applications of metaheuristics inspired nature in combinatorial logistic problem
The focus of the paper is on a logistics problem, related to the transport of goods, which can be applied in practice, for example in postal or courier services. Two mathematical models are presented as problems occurring in a logistics network. Models are solved by CPLEX. For large problems, two nature-inspired methods have been proposed: evolutionary algorithm and artificial bee colony algorithm. Authors have presented a package of combinatorial optimization problems, their models and solution methods for different sizes of each problem.
The paper is generally good in the sense of organization. After the first revision, the paper’s logic and contribution is clear to me. So, my comments in this revision are more about enhancing the quality of the paper to be more fit for the logistic study literature. Please implement the following comments.
- To begin with, I believe that the topic of the paper is general. have authors studied all combinatorial logistic problems in this paper. The list of combinatorial logistic problem is too long, and so I do not think authors have studied all of them. It would be great if authors can replace “combinatorial logistic problem” with the specific title of the problem studied in this paper.
- Authors should use one of American or British English (e.g., levelized, optimization, summarize, minimize, customize, normalize or levelised, optimization, summarise, minimise, customise, normalise) in the paper, not both.
- Subsection 1.1 and 1.2 do not have a lot to say as an independent subsections. They can be merged into Section 1 for the sake of simplicity. Specifically, Subsection 1.2 is only a paragraph that describe the roadmap of the paper. It is not common to consider it as an independent subsection.
- In the model of Page 5, there should be a “S.t.” between Equations 1 and 2 to clarify (1) is objective function and the rest from (2) to (6) are constraints of the model. In addition, a constraint should be added to the end of the model to show how domains of variables are sets. This shows which one of variables are binary and which one of them are nonnegative continuios.
- Only as a suggestion, it is good to have rigid definitions and proper citations for important terms. An example is “VRP” with the following definition “What is the optimal set of routes for a fleet of vehicles to traverse in order to deliver to a given set of customers [a]”. Moreover, the definition of another important logistics problem, TSP, should be added to the paragraph: “what is the shortest possible route for a single vehicle to visit each customer exactly once and returns to the source [b]” [a] vehicle routing problem for reverse logistics of end-of-life vehicles, Waste Management, vol. 120, 209–220 [b] a transformation technique for the clustered generalized traveling salesman problem with applications to logistics, European Journal of Operational Research, vol. 285, 444-457
- Inconsistency: please use one of “Fig.” or “Figure” in the body of the paper, not both (e.g., see Pages 8). I have not checked this inconsistency in the paper entirely. I ask authors to polish the paper once and be sure all typos are corrected.
- The title of some papers in the reference list is lowercase, whereas the others are Capitalized each Word. This is also a case of inconsistency. Please make all of them lowercase for the sake of simplicity.
- Other errors:
Page 2: delivery operations often boils down --> delivery operations often boil down
Page 7: witch is the stop --> which is the stop
Page 8: Such observation is strengthen by --> Such observation is strengthened by
Page 8: column T in tables 1 and 3 --> column T in Tables 1 and 3
Author Response
Dear Sir or Madam,
We are very grateful to the reviewer for taking the time to review the manuscript and providing comments that give us a chance to improve its quality. Newly added or substantially modified portions of the revised manuscripts are marked in yellow. Minor edits (like typo corrections) are not marked. Responses to specific suggestions are provided in the attached pdf file.
Best regards,
Stanislaw Kozdrowski

Reviewer 2 Report
line47-48: np-complete -> np-hard
line56: efficient exact algorithms is not rather unlike
line134: np-complete -> np-hard
Section 2: describe WER, PER, D3, D2, D1, D0, DEL (line 160, 161)
model (1)-(6): nonnegativity constraints are missing
model (7)-(9): nonnegativity constraints are missing
Note that only the y_ta variables are integers, so I think the model should be very easy to solve by CPLEX.
The article does not describe the parameter calibration of the proposed metaheuristics (EA/ABC)
One would expect the computation time associated with the heuristics to be polynomial, but it has exponential behavior according to the tables and figures.
the instances are not available, therefore, it is not possible to repeat the experiment
Author Response
Dear Sir or Madam,
We are very grateful to the reviewer for taking the time to review the manuscript and providing comments that give us a chance to improve its quality. Newly added or substantially modified portions of the revised manuscripts are marked in yellow. Minor edits (like typo corrections) are not marked. Responses to specific suggestions are provided in the attached pdf file.
Best regards,
Stanislaw Kozdrowski.

Round 2
Reviewer 2 Report
- the authors improved the scientific articles.
- It is recommended for future articles to work more on the MIP model, to improve the linear relaxation (and the lower bound)
- It is also recommended that heuristic/metaheuristic algorithms have a polynomial computation time, since it is expected that very large instances can be solved.
Author Response
Dear Sir or Madam,
Thank you for your valuable comments. The responses are attached as a pdf file.
Best regards,
Stanislaw Kozdrowski.
